# Ethylene Modulates Rice Root Plasticity under Abiotic Stresses

**DOI:** 10.3390/plants13030432

**Published:** 2024-02-01

**Authors:** Hua Qin, Minggang Xiao, Yuxiang Li, Rongfeng Huang

**Affiliations:** 1Biotechnology Research Institute, Chinese Academy of Agricultural Sciences, Beijing 100081, China; liyx0929@163.com (Y.L.); rfhuang@caas.cn (R.H.); 2National Key Facility of Crop Gene Resources and Genetic Improvement, Beijing 100081, China; 3Biotechnology Research Institute, Heilongjiang Academy of Agricultural Sciences, Harbin 150028, China; xiaoyang8076@163.com

**Keywords:** root development, ethylene, abiotic stress, rice

## Abstract

Plants live in constantly changing environments that are often unfavorable or stressful. Root development strongly affects plant growth and productivity, and the developmental plasticity of roots helps plants to survive under abiotic stress conditions. This review summarizes the progress being made in understanding the regulation of the phtyohormone ethylene in rice root development in response to abiotic stresses, highlighting the complexity associated with the integration of ethylene synthesis and signaling in root development under adverse environments. Understanding the molecular mechanisms of ethylene in regulating root architecture and response to environmental signals can contribute to the genetic improvement of crop root systems, enhancing their adaptation to stressful environmental conditions.

## 1. Introduction

As a staple food for more than half of the world’s population, the demand for rice (*Oryza sativa* L.) production is increasing with the continuous increase in the world population; however, the production of rice is threatened by a variety of abiotic stresses, including salinity, heat, drought, and submergence, and projected climate change aggravates the effects of these abiotic stresses [1,2,3]. Since plants cannot avoid abiotic stress by moving, they evolved elaborate mechanisms to cope with diverse environmental stresses, and phenotypic plasticity is thought to enable plants to cope with or even take advantage of environmental heterogeneity [4,5,6]. Elucidating the molecular mechanisms of how plants sense stress signals and adapt to adverse environments is critical for agricultural productivity and, therefore, for global food security.

Roots are important to plants for the anchorage, uptake, storage, and transport of minerals and water. As the belowground organ of the plant, roots serve as the major interface between the plant and various biotic and abiotic factors in the soil environment [7,8]. Upon exposure to stress, roots are the first organ to perceive these stress signals and adapt their architecture accordingly, thus helping plants survive under stressful conditions [8]. Therefore, root development strongly affects plant growth and productivity, and the optimization of root development is expected to be crucial for enabling the next Green Revolution [9,10].

The growth and development of roots are controlled by endogenous genetic programs and exogenous signals, and the gaseous phytohormone ethylene plays an important role in this process [11,12,13,14]. Ethylene is synthesized from S-adenosylmethionine (SAM), which is converted to 1-aminocyclopropane-1-carboxylate (ACC) by ACC synthase (ACS). ACC is then converted to ethylene by ACC oxidase (ACO) [15]. Once ethylene is synthesized, it is perceived by a family of receptors that negatively regulate ethylene responses. The signal is transmitted via a linear pathway, consisting of the negative regulator CONSTITUTIVE TRIPLERESPONSE 1 (CTR1), the positive regulator ETHYLENE INSENSITIVE 2 (EIN2), the master transcription factors EIN3 and EIN3-LIKE1 (EIL1), and the ethylene response factors (ERFs) [16]. Besides the conserved components of ethylene signaling, several novel regulators and mechanisms of the rice ethylene signaling pathway were recently identified [17,18,19,20]. Histidine kinase (HK) MAO HUZI 1 (MHZ1) autophosphorylates at a conserved histidine residue and can transfer the phosphoryl signal to the response regulator (RR) OsRR21 via the phosphotransfer proteins OsAHP1/2 and interacts with ethylene receptors to regulate root growth in rice [17]. Upon ethylene treatment, ethylene receptors released the repression effect on MHZ1 and the phosphor-relay pathway is activated. The MHZ1-mediated phosphor-relay pathway and the OsEIN2-mediated pathway function together to regulate a subset of downstream genes to modulate root growth [17]. MHZ3, an endoplasmic reticulum (ER) membrane-localized protein, interacts with the N-terminal Nramp-like domain of OsEIN2 to stabilize it by inhibiting ubiquitination and proteasome-mediated degradation [18]. MHZ9, a glycine–tyrosine–phenylalanine (GYF) domain-containing protein, interacts with OsEIN2-CEND in P-body and directly binds to the OsEBF1/2 mRNAs for translational repression, leading to the accumulation of OsEIL1 to activate the downstream signaling [19]. MHZ11, an ER membrane GDSL-family lipase with acyl-hydrolyzing activity, facilitates ethylene-induced conformational changes in ethylene receptors to suppress OsCTR2 phosphorylation by affecting the membrane sterol homeostasis [20]. In rice, ethylene treatment inhibited primary root elongation, promoted the emergence and growth of adventitious roots from an aerial nodal, and caused root swelling [12,21,22], implying the diverse roles of ethylene in rice root development.

In addition to regulating root growth and development, ethylene also plays a role in plant response to abiotic stresses [23,24]. In rice, ethylene biosynthesis is induced by multiple abiotic stresses, such as drought, salinity, and submergence [13,25]. Enhancing ethylene biosynthesis increased resistance to drought stress, whereas disrupting ethylene transduction increased the salt tolerance of rice [26,27], revealing the distinct role of ethylene in abiotic stresses and suggesting that precise manipulation of ethylene biosynthesis and signaling will contribute to improving abiotic stress tolerance in rice.

Based on the crucial role of ethylene in root development and stress tolerance, this review mainly focuses on recent advances in research on ethylene regulation of root growth and development in response to abiotic stresses in rice, which will provide readers with information on use for their own research and new breeding strategies for stress-resilient rice varieties with stable yields, even in a challenging environment.

## 2. Improving Root System Contributes to Increase Rice Yield and Stress Resistance

As a monocot model plant, the rice root system consists of the primary root, adventitious roots (ARs), also called crown roots (CRs), and lateral roots (LRs) [28] (Figure 1A). The embryonic primary root in rice plays an important role in only a transient period after germination and then disappears when ARs take over; thus, the ARs are the main components of the mature root system in rice [29]. Root system architecture (RSA) describes the spatial arrangement of root components within the soil and determines the plant’s exploration of the soil. The RSA of crops can affect their production, particularly in abiotic stress conditions [30,31]. RSA is determined by a number of parameters, including root growth rate, the rate at which LRs and ARs are formed, and the angles at which LRs and ARs grow with respect to the gravity vector [32].

Accumulating studies showed that root development is associated with rice yield and tolerance to abiotic stresses [7]. The introgression of quantitative trait loci (QTL) associated with root traits increased root length, root thickness, and yield in upland rice [33]. Using mutant and QTL analyses, many genes related to the development of the rice root system were identified [34] (Table 1). *CROWN ROOTLESS1/ADVENTITIOUS ROOTLESS1* (*CRL1/ARL1*) was the first identified gene controlling AR formation. The *crl1* mutant completely lacks ARs, and the number of LRs on primary root was reduced. Moreover, no normal tillers were observed in *arl1* mutant plants because the plants died due to the lack of ARs [35,36]. The *WUSCHEL-related Homeobox* (*WOX*) gene *WOX11*, specifically expressed in the emerging AR meristem, acts synergistically with CRL1 to promote AR development in rice [37,38]. The overexpression of *WOX11* increases rice drought resistance by modulating root hair formation and RSA development [39]. *DEEPER ROOTING 1* (*DRO1*) and *SOIL SURFACE ROOTING 1* (*qSOR1*), two QTLs associated with root growth angle, increasing the expression of *DRO1,* resulting in a more downward direction of root grows, thereby increasing root depth to avoid drought and maintained high yield performance under drought conditions [31], whereas the loss-of-function allele *qsor1* resulted in roots that developed on the soil surface and enabled plants to avoid the reducing stress found in saline paddy soils and, consequently, increased yields [30]. R2R3-type MYB family transcription factor ROBUST ROOT SYSTEM 1 (RRS1) negatively regulates root development by modulating auxin signaling transduction. Knockout of *RRS1* not only promotes root development but also improves drought resistance without loss of grain yield [40]. All these studies suggest that improving RSA is beneficial to increasing rice yield and tolerance to abiotic stress.

Several studies showed that targeting roots, rather than altering responses on the entire plant, emerged as an alternative strategy for improved abiotic stress tolerance and yield in rice [7]. The root-specific overexpression of *OsERF71* alters radical root growth and increases drought resistance and yield under drought stress. Moreover, root-specific overexpression was more effective in conferring drought resistance at the reproductive stage, resulting in an increase in grain yield by 23% to 42% over wild-type plants under drought conditions [56]. Similarly, the root-specific overexpression of *OsNAC5*, *OsNAC6*, *OsNAC9*, and *OsNAC10* promoted the root diameter and stress tolerance in rice, and the grain yield of the overexpressors was higher than that of wild-type controls under both normal and stress conditions [57,58,59,60], suggesting that the modulation of root development might be the key to improving stress tolerance without yield penalties.

## 3. Ethylene Confers Diverse Roles in Rice Root Development

The phytohormone ethylene plays a profound role in plant growth and development [11,61]. In rice, ethylene promotes the coleoptile growth of etiolated rice seedlings but inhibits root elongation [22]. Through screening and investigating rice mutants with defective ethylene response, the core components of rice ethylene signaling, OsEIN2 and OsEIL1, were identified. The mutation of *OsEIN2* or *OsEIL1* results in complete ethylene insensitivity in primary root growth [22,27], indicating that ethylene signaling components are important for ethylene-inhibited root growth. In addition, some auxin and abscisic acid (ABA) biosynthesis mutants, such as *mhz4*, *mhz5*, *mhz10*, and *rice ethylene insensitivity 7* (*rein7*), were also identified [47,48,49,51], suggesting that auxin and ABA are required for ethylene-inhibited primary root elongation.

Root elongation is determined by the cell division in the root apical meristem (RAM) and the elongation of cells that leave the RAM [62] (Figure 1B). Increasing evidence indicates that ethylene inhibits root elongation by restricting epidermal cell elongation, and this effect is mainly through crosstalk with auxin [63,64]. Since continuous root growth is sustained by cell division and cell differentiation within the RAM [65], mutant rice with defects in the division and survival of RAM cells show severe defects in root growth [66]. Numerous studies showed that ethylene plays an important role in controlling meristem activity by restricting cell proliferation in RAM [67,68]. Ethylene treatment significantly reduced RAM size and the cell number in RAM. Similarly, the overexpression of *OsEIN2* and *OsEIL1* resulted in a smaller root meristem with reduced cell number and exhibited a much shorter and twisted primary root [68], suggesting that ethylene inhibits cell proliferation in root meristem to restrict rice root elongation. Further investigation unraveled that ethylene promotes gibberellin (GA) metabolism in roots, which further controls cell proliferation in the root meristem and primary root elongation [68]. In addition to inhibiting root elongation, ethylene also functions in the horizontal growth of roots. In rice, ethylene treatment increases root diameter by increasing the radial expansion of cortical cells, and this process is dependent on ABA [12,46]. All these studies indicate that ethylene has biphasic effects on root growth via interaction with other phytohormones.

Apart from the primary root, ethylene also plays an important role in AR initiation and development in rice [69]. Exogenous ethylene treatment promoted the emergence and growth of ARs, and dark-induced AR growth was suppressed by 1-methylcyclopropene (1-MCP), an inhibitor of ethylene perception [21,70], indicating that ethylene positively regulates AR growth and development in rice. AR development is closely associated with auxin since disruptions in auxin biosynthesis, transport, and signaling lead to abnormal AR development in rice [34]. Ethylene triggers auxin accumulation in roots through OsEIL1-mediated activation of auxin biosynthesis genes *MHZ10* and *REIN7*, thereby disrupting auxin signaling transduction by promoting SOR1-dependent degradation of Aux/IAA protein [49,51,71]. Blocking auxin transport with 1-N-naphthylphthalamic acid (NPA) attenuated ethylene-promoted AR growth in rice [21], suggesting that ethylene directs auxin to control AR development. OsERF3 and OsWOX11 are positive regulators of AR development in rice; they cooperatively regulate cytokinin signaling to control AR development [29]. Moreover, OsERF3 negatively regulates ethylene biosynthesis and OsWOX11 functions downstream of auxin in AR development [26,72], suggesting a potential function of OsERF3 and OsWOX11 in ethylene- and auxin-modulated AR development. Further research should focus on the ethylene effect on AR development and illuminating the underlying molecular mechanism.

LRs are the major components involved in the absorption of nutrients and the interaction with the surrounding soil environment [34]. In rice, treatment of seedlings with ethylene precursor ACC had no significant effect on the LR number but significantly inhibited LR elongation. Correspondingly, the number and length of LRs were increased in seedlings treated with ethylene biosynthesis inhibitor L-α-(2-aminoethoxyvinyl)-glycine (AVG) [73]. OsZHD2, which encodes a zinc-finger homeobox (ZF-HD) protein, positively regulates LR development by modulating ethylene biosynthesis [74], suggesting that ethylene also participates in the regulation of LR development in rice. However, the molecular mechanisms of ethylene in rice LR development should be further investigated.

## 4. Ethylene Regulates Root Development under Abiotic Stresses

As the belowground organ of the plant, roots encounter varying environmental conditions and respond by altering their growth, and the plant hormone ethylene plays an important role in this process [12,43,52].

### 4.1. Salinity

Soil salinity is one of the major abiotic stresses that limit food crop growth and yield worldwide. Rice is considered to be a salt-susceptible species, and its salt tolerance greatly depends on growth stages, organ types, and cultivars. Significantly, the roots of young rice seedlings are highly salt-sensitive organs that limit plant growth, even under mild soil salinity conditions [75]. Accumulating studies showed that salt stress inhibits root elongation and promotes root swelling, and ethylene participates in this process [13,52,76]. In rice, mutations in ethylene signaling components OsEIN2 and OsEIL1 lead to longer primary roots and enhanced salt tolerance [22,27]. AVG treatment significantly reversed salt-inhibited root growth in rice [13]. Moreover, salt stress induces ethylene biosynthesis in rice roots and jasmonate (JA) functions downstream of ethylene to inhibit root growth under salt stress [13]. DNA-BINDING WITH ONE FINGER 15 (OsDOF15), a salt-responsive transcription factor, positively regulates root growth under salt stress by modulating ethylene biosynthesis [52]. SALT INTOLERANCE1 (OsSIT1), a lectin receptor-like kinase expressed mainly in root epidermal cells, positively regulates salt tolerance by activating MITOGEN-ACTIVATED PROTEIN KINASE 3/6 (MPK3/6), which promotes ethylene and ROS production [53], suggesting that OsDOF15 and OsSIT1 may act as a link between salt and ethylene biosynthesis in salt-inhibited root growth in rice.

### 4.2. Drought

Drought occurs when crops are subjected to insufficient soil moisture to meet their demands. Drought directly hampers root growth and development and induces parsimonious root architecture with fewer LRs and a generally deeper rooting structure [8]. Deeper rooting allows for efficient water capture and, thereby, ameliorates drought stress. As a semi-aquatic plant, rice is particularly susceptible to drought stress. Rice plants with deep roots are more tolerant to drought stress and maintain productivity in such circumstances [31]. Ethylene was reported to participate in drought tolerance. Drought stress induces ethylene biosynthesis in rice, and reduced ethylene biosynthesis leads to drought sensitivity in rice [25]. Ethylene overproducer 1-like gene (OsETOL1), a homolog of *Arabidopsis* ETO1, encodes a putative E3 ubiquitin ligase and modulates drought tolerance by interacting with OsACS2 to regulate ethylene biosynthesis in rice [25]. The overexpression of *ACIREDUCTONE DIOXYGENASE 1* (*OsARD1*) enhances ethylene biosynthesis and improves root growth under drought stress, thus increasing the drought tolerance of rice seedlings [54]. OsERF3 positively regulates AR development in rice; the overexpression of *OsERF3* decreases drought tolerance through repressing ethylene biosynthesis [26,29]. Drought-responsive ERF gene 1 (OsDERF1) and OsERF109, upstream regulators of OsERF3, negatively regulate drought tolerance by activating OsERF3-mediated ethylene biosynthesis [45,77]. All these studies indicate that enhanced ethylene biosynthesis could improve drought tolerance in rice. Considering that ethylene plays an important role in root development, whether these genes regulating drought tolerance in rice are dependent on ethylene-modulated root development should be further investigated.

### 4.3. Soil Compaction

Soil compaction represents a major challenge for modern agriculture. Compacted soil layers constrain crop productivity by restricting root growth and exploration in deeper soil domains, which in turn limits access to nutrients and water [32]. The various root responses to soil compaction were reported in rice, such as shortened root length and increased root diameter, and ethylene acts as an early warning signal for roots to avoid compacted soils [12,78]. Compacted soil restricts the outward diffusion of ethylene through smaller air-filled soil pores, leading to the accumulation of OsEIL1 in the roots, which further activates auxin biosynthesis in epidermal cells and ABA biosynthesis in cortical cells, thereby inhibiting epidermal cell elongation and promoting the radial expansion of cortical cells, ultimately resulting in short and swollen roots [12,46]. Rice mutants deficient in ethylene signaling, auxin biosynthesis and transport, and ABA biosynthesis exhibited better root penetration ability in harder soil [12,46,50]. In addition, circumnutation is a preferable strategy for roots to navigate an environment containing heterogeneities such as rocks. Recent studies showed that ethylene activates *OsHK1*, a positive regulator of ethylene signaling and functions downstream of ethylene receptors, to promote root circumnutation and inhibit root elongation in rice [17,43]. The role of ethylene in root circumnutation and root elongation facilitates root growth past obstacles but inhibits root elongation in harder soils. Therefore, the precise manipulation of ethylene actions would increase the performance of crops under specific agronomical conditions.

### 4.4. Hypoxic Conditions

Rice lives in a water-saturated environment most of the time during its life cycle; this means that rice roots grow under hypoxic conditions. To adapt to hypoxic conditions, rice form lysigenous aerenchyma in roots as a result of cell death and lysis of the root cortical cells [79] (Figure 1C). Hypoxic conditions promote ethylene accumulation in roots due to ethylene biosynthesis and its low diffusion rate to the rhizosphere [80,81]. Upon accumulation, ethylene stimulates the programmed cell death that occurs during lysigenous aerenchyma formation [82]. Studies in rice showed that NADPH oxidase respiratory burst oxidase homolog (RBOH) H-mediated reactive oxygen species (ROS) production is essential for ethylene-dependent aerenchyma formation in rice roots under oxygen-deficient conditions [55]. Ethylene perception inhibitor 1-MCP treatment suppresses the expression of *RBOHH* and aerenchyma formation in rice roots under oxygen-deficient conditions [55], suggesting that *RBOHH* expression is regulated through ethylene signaling. In addition, ethylene signaling transcription factor OsEIL1 directly binds to the promoters of *OsRBOHA* and *OsRBOHB* to activate their expression, leading to the accumulation of ROS and the increase in disease resistance [83]. Recent studies in rice showed that auxin signaling act downstream of the systemic ROS signaling to promote LR development in response to asymmetric stress of heavy metals and salt, and AUX/IAA-mediated auxin signaling contributes to ethylene-dependent inducible aerenchyma formation [84,85]. These data suggest that ethylene employs an RBOH-ROS-auxin signaling cascade to regulate the adaption of rice roots to stressful environments such as hypoxic conditions.

### 4.5. Nutrients

The distribution of nutrients in the soil also affects root development in rice [86]. Soil phosphate and nitrogen are the major constraints for crop productivity, and plants alter their RSA to increase phosphate and nitrogen acquisition at minimum cost [86,87]. Phosphate is an essential macro-element for plant growth accumulated in the topsoil. Shallow root systems that improve access to phosphate are formed through increasing AR growth angle, LR number, and LR length in rice under low-phosphate conditions [88]. Several studies demonstrated that ethylene functions in modulating RSA in low-phosphate conditions [89,90]. Phosphate starvation induces the expression of ethylene biosynthesis genes and ethylene production in rice roots [89,91]. ACC accentuates the remodeling of root architecture in response to phosphorus, and mutation in *OsACS1* or *OsACS2* results in reduced LR elongation under phosphorus-deficient conditions [91], suggesting that the regulation of ethylene biosynthesis via *OsACS* genes is probably one of the strategies that rice plants use to cope with phosphorus deficient stress conditions. Unlike phosphorus, nitrogen is another essential macro-element for plant growth accumulated in the subsoil, and nitrate (NO_3_^−^) and ammonium (NH_4_^+^) are the primary sources of nitrogen for crop plants. Both nitrate and ammonium treatment stimulates AR formation in rice, whereas ammonium represses and nitrate promotes the elongation of ARs [87]. The knockout of *OsEIL1* results in hypersensitivity to ammonium in shoots and roots, and OsEIL1 directly activates GDP-mannose pyrophosphorylase (VTC1) *OsVTC1-3* to maintain the stability of protein N-glycosylation to constrain ammonium efflux in rice [41]. All these results suggest that ethylene is an important cellular signal to mediate root system architecture adaptation to external phosphate and nitrogen status.

### 4.6. Heavy Metals

Heavy metals, such as cadmium, copper, and lead, cause adverse effects on the root system by inhibiting root elongation, repressing LR and AR initiation, and inducing root tip radial swelling [92]. Rice remodeling root architecture to avoid asymmetric heavy metal by rapidly proliferating LRs in the stress-free area and a root tip-specific burst of ROS-directed auxin signaling is required for this process [85]. Moreover, blocking the ethylene signal resulted in ROS accumulation and the inhibition of root growth under Cd stress. Conversely, ethylene signal enhancement by *OsEIN2* overexpression displayed higher Cd tolerance [42]. These results suggest an ethylene–ROS–auxin signaling cascade that enables rice roots to avoid heavy metal stress.

## 5. Conclusions and Perspectives

As sessile organisms, plants are continuously impacted by changes in the environment. To adapt to adverse situations, plants evolved well-developed mechanisms that help to alleviate environmental stresses. Roots are the primary target site for the perception of stress signals and show high developmental plasticity to cope with environmental challenges [8]. Thus, improving root growth and development could be a key strategy for the development of crops with improved stress tolerance and yield, minimizing or avoiding the penalties of growth-defense trade-offs. Ethylene is a well-known plant growth regulator that mediates adaptations to environmental conditions [23,24,93]. Numerous studies showed that abiotic stresses, including salt, drought, hypoxic, and soil compaction, activate the expression of ethylene biosynthesis genes to promote ethylene accumulation in roots, which further employs auxin, ABA, JA, and GA as downstream signals to modulate root development (Figure 2), thereby helping rice to maintain the optimal growth and yield under adverse conditions.

The role of ethylene in rice root development was extensively studied [11,94]. However, most studies on ethylene functions were focused on primary roots, while the molecular mechanism of ethylene in AR and LR development is largely unknown. Since ARs and LRs constitute the major root system of rice, further studies should focus on elucidating the regulatory network of ethylene in AR and LR development. In addition, ethylene inhibits root elongation, promotes root swelling, and enhances AR and LR initiation [12,21,74], leading to the formation of shallow roots. Shallow rooting is advantageous for the acquisition of nutrients such as phosphorus from the topsoil and enables rice to avoid the reducing stresses in saline paddy soils, whereas deep rooting is favorable for the acquisition of water and nitrogen from the subsoil and increases rice yield under drought conditions [30,31,95]. Studies showed that ethylene acts as an early warning signal for roots to avoid adverse environments [12,52]. Changes to the ethylene signaling pathway affect rice grain filling and grain size, ultimately affecting yield [61]. Thus, the molecular mechanism of how plants perceive external changes and translate cues into adaptive responses by modulating ethylene biosynthesis and signaling should be further investigated, and the identification of the favorable alleles of ethylene biosynthesis and signaling components would contribute to cultivating elite rice varieties that could achieve optimal growth and higher grain production under both normal and adverse conditions.

Breeding plants with important root traits is promising for developing crops with increased yield and resistance to adverse environments, and optimization or modulation of root development may enable the next Green Revolution [9]. Although a member of genes involved in rice root development was identified [34] (Table 1), only a few of them were used in rice breeding. A more thorough understanding of the key genes associated with RSA and their regulation should enable breeders to generate cultivars with enhanced root systems using marker-assisted selection in the future.

As a below-ground organ of plants, roots are encased in soil and cannot be visualized. Thus, root phenotyping is a challenging task in comparison to shoot. Until now, studies on rice root systems are mainly carried out in seedlings grown in aqueous solution or medium, which is distinct from root systems grown in the paddy field. With the development of technology, various root phenotyping methods were established, such as X-ray computed tomography (CT) imaging [96]. Although CT could monitor the root morphology in the soil, it requires specific equipment and a high expense for image collection. Moreover, restricted soil conditions and plants in their early growth stages are required to collect three-dimensional (3D) images. Rice is a semi-aquatic plant that grows in a water-saturated environment for most of its life cycle; this makes it more difficult to monitor its roots. The development of simple, fast, and low-cost detection methods will greatly accelerate the research of rice root development.

## Figures and Tables

**Figure 1 plants-13-00432-f001:**
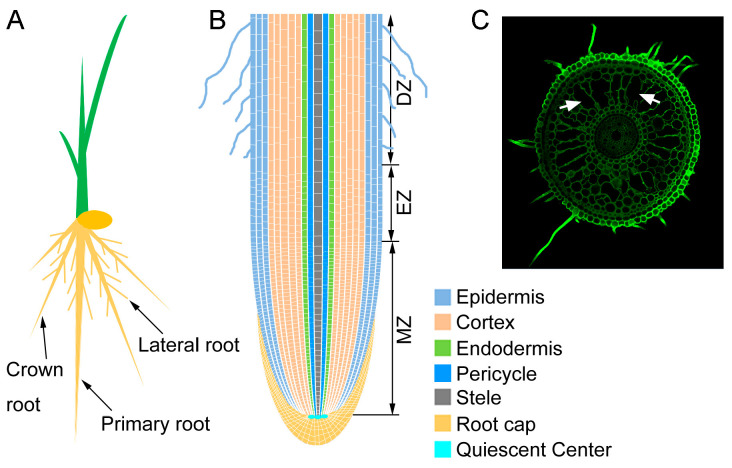
Anatomy of rice roots. (**A**) A schematic of rice root system composed of primary, crown, and lateral roots. (**B**) A schematic of longitudinal section of rice root. Colors in the diagram indicate corresponding cell types. MZ: meristematic zone; EZ: elongation zone; DZ: differentiation zone or maturation zone. (**C**) A radial anatomy of rice roots was visualized by transverse section. Arrows indicate aerenchyma, respectively.

**Figure 2 plants-13-00432-f002:**
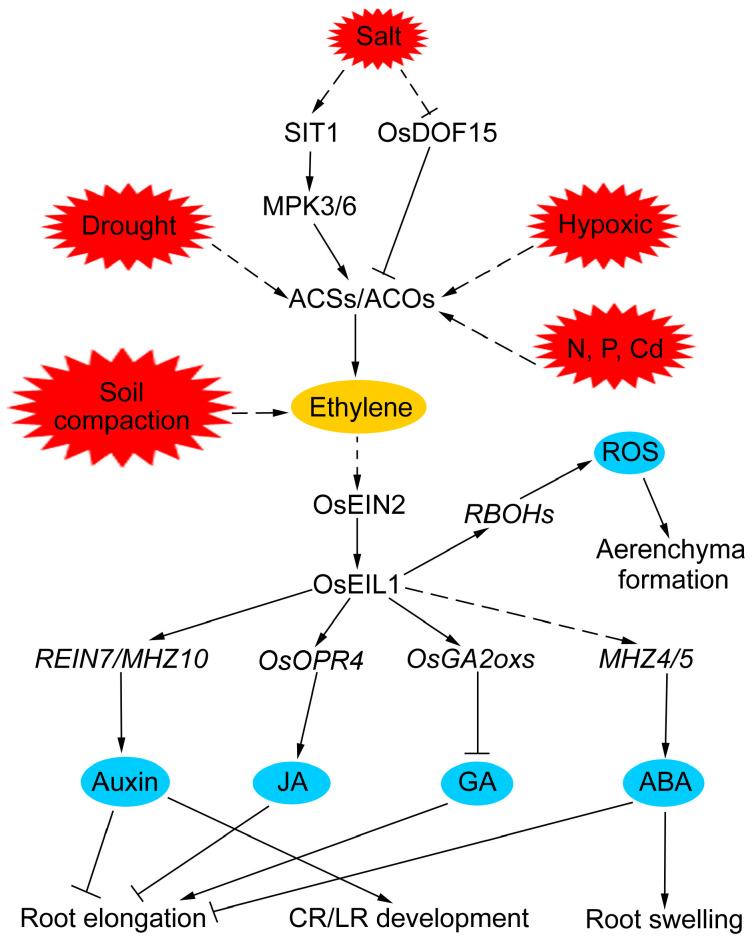
The molecular regulatory mechanisms of ethylene in rice root development in response to adverse environments. Arrows represent positive regulatory actions. Lines ending in a flat head indicate a negative regulatory action. The solid lines indicate direct interactions, and the dashed lines indicate indirect interactions.

**Table 1 plants-13-00432-t001:** Ethylene-related genes involved in root development and abiotic stress tolerance.

Gene Name	Gene Function	Reference
*OsEIL1*	Regulates ethylene response of roots and salt tolerance in rice, maintains rice growth under NH_4_^+^ by stabilizing protein N-glycosylation and reducing root NH_4_^+^ efflux	[27,41]
*OsEIN2*	Regulates ethylene response of roots and salt tolerance in rice, positively regulates Cd tolerance in rice roots	[22,27,42]
*MHZ1/OsHK1*	Positively modulates root ethylene responses and root circumnutation	[17,43]
*OsWOX11*	promotes AR development by modulating cell proliferation in AR meristem, enhances drought tolerance by controlling root hair formation and root system development	[37,39,44]
*OsERF3*	Regulates AR development through interacting with OsWOX11, regulates drought tolerance in rice through modulating ethylene biosynthesis	[26,29]
*OsDERF1*	Negatively modulates ethylene synthesis and drought tolerance in rice by directly activating *OsERF3* and *OsAP2-39*	[45]
*MHZ4*	Regulates ethylene response of roots and the ability of root to penetrate compacted soil	[46,47]
*MHZ5*	Regulates ethylene response of roots and the ability of root to penetrate compacted soil	[46,48]
*MHZ10/TAA1*	Regulates ethylene response of roots and the ability of root to penetrate compacted soil	[49,50]
*REIN7/YUC8*	Regulates ethylene response of roots and the ability of root to penetrate compacted soil	[46,51]
*OsDOF15*	Regulates root growth under salt stress by modulating ethylene biosynthesis	[52]
*OsSIT1*	Specially expressed in root epidermal cells, regulates salt tolerance by promoting ethylene and ROS production in rice	[53]
*OsARD1*	Improves root growth under drought stress and increases submergence, salt, and drought tolerance in rice	[54]
*RBOHH*	Regulates aerenchyma formation in rice roots under oxygen-deficient conditions	[55]

## Data Availability

All the data in this study are available within this article.

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
