# Peer review of "Ethylene Modulates Rice Root Plasticity under Abiotic Stresses"

_plants, 2024, doi:10.3390/plants13030432_

Round 1

Reviewer 1 Report

Comments and Suggestions for Authors

Qin et al highlights the role of ethylene in regulating root growth and response to various stresses. This manuscript is well written. I just have some mirror points:

1). Line 12: summarize should be summarizes

2). The author should short the part of “2. Improving root system contributes to increase rice yield and stress resistance”, since this part is not related to the role of ethylene.

3). The author should summarize the rice ethylene signaling pathway in the Introduction part.

4). The author should delete these genes which are not involved in ethylene signaling in Table 1.

5). Line 176-179, the author should delete this part.

Comments on the Quality of English Language

This manuscript is well written.

Author Response

Qin et al highlights the role of ethylene in regulating root growth and response to various stresses. This manuscript is well written.

Response: Thanks very much for your positive comments.

I just have some mirror points:

1). Line 12: summarize should be summarizes

Response: We have corrected it in the revised manuscript.

2). The author should short the part of “2. Improving root system contributes to increase rice yield and stress resistance”, since this part is not related to the role of ethylene.

Response: Thank you for your suggestion. We removed some contents of this section in the revised manuscript.

3). The author should summarize the rice ethylene signaling pathway in the Introduction part.

Response: Thank you for your constructive suggestion. Besides the conserved components of ethylene signaling, several novel regulators and mechanisms of the rice ethylene signaling pathway have recently been identified [17-20]. Histidine kinase (HK) MAO HUZI 1 (MHZ1), autophosphorylates at a conserved histidine residue and can transfer the phosphoryl signal to the response regulator (RR) OsRR21 via the phosphotransfer proteins OsAHP1/2, interacts with ethylene receptors to regulate root growth in rice [17]. Upon ethylene treatment, ethylene receptors released the repression effect on MHZ1 and the phosphor-relay pathway is activated. The MHZ1-mediated phosphor-relay pathway and the OsEIN2-mediated pathway function together to regulate a subset of downstream genes to modulate root growth [17]. MHZ3, an endoplasmic reticulum (ER) membrane-localized protein, interacts with the N-terminal Nramp-like domain of OsEIN2 to stabilize it by inhibiting ubiquitination and proteasome-mediated degradation [18]. MHZ9, a glycine-tyrosine-phenylalanine (GYF) domain-containing protein, interacts with OsEIN2-CEND in P-body and directly binds to the OsEBF1/2 mRNAs for translational repression, leading to the accumulation of OsEIL1 to activate the downstream signaling [19]. MHZ11, an ER membrane GDSL-family lipase with acyl-hydrolyzing activity, facilitates ethylene-induced conformational changes in ethylene receptors to suppress OsCTR2 phosphorylation by affecting the membrane sterol homeostasis [20]. We have added these in the Introduction part in the revised manuscript.

4). The author should delete these genes which are not involved in ethylene signaling in Table 1.

Response: We have done as suggest.

5). Line 176-179, the author should delete this part.

Response: We deleted this part in the revised manuscript.

Reviewer 2 Report

Comments and Suggestions for Authors

The review manuscript “Ethylene facilitates rice root development under abiotic stress” by Qin et al. is a well-written, easy to read manuscript. It covers several aspects of the importance of ethylene (and its interaction with other hormones) to regulate root system architecture plasticity to different abiotic stresses in rice. It is a nice work to start to get into root development in rice mediated by ethylene.

Title: I think the title can be a little misleading…sometimes ethylene may not facilitate! I would suggest something like “Ethylene is a key regulator of root plasticity under abiotic stress in rice” or something around this. It is just a suggestion.

I would also ask the authors to normalize the adventitious roots (ARs)/crown roots (CRs) acronym throughout the manuscript. Sometimes the authors refer to adventious roots (ARs) but in other moments refer to CRs. Wouldn’t it be better to use only one acronym? Since AR and CR are the same, using different acronyms could lead to confusion.

Line 175-176/180: these sentences seem a bit contradictory. In one sentence, ethylene promotes LR formation and elongation, but in the other ACC had no significant effect on LR number but inhibited LR elongation. Could you give an explanation?

Line 246-250: Again, it seems a bit contradictory… If ethylene mutants have a better penetration in compact soil, circumnutation should be important to it. If HK1 is a positive regulator of ethylene, how can this be useful for penetration in compact soils and root elongation?

Author Response

The review manuscript “Ethylene facilitates rice root development under abiotic stress” by Qin et al. is a well-written, easy to read manuscript. It covers several aspects of the importance of ethylene (and its interaction with other hormones) to regulate root system architecture plasticity to different abiotic stresses in rice. It is a nice work to start to get into root development in rice mediated by ethylene.

Response: We appreciate the positive comments by the reviewer regarding quality of the manuscript.

Title: I think the title can be a little misleading…sometimes ethylene may not facilitate! I would suggest something like “Ethylene is a key regulator of root plasticity under abiotic stress in rice” or something around this. It is just a suggestion.

Response: Thank you for your suggestion. We changed the tittle to “Ethylene modulates rice root plasticity under abiotic stresses” in the revised manuscript.

I would also ask the authors to normalize the adventitious roots (ARs)/crown roots (CRs) acronym throughout the manuscript. Sometimes the authors refer to adventious roots (ARs) but in other moments refer to CRs. Wouldn’t it be better to use only one acronym? Since AR and CR are the same, using different acronyms could lead to confusion.

Response: We used adventitious roots (ARs) throughout the revised manuscript.

Line 175-176/180: these sentences seem a bit contradictory. In one sentence, ethylene promotes LR formation and elongation, but in the other ACC had no significant effect on LR number but inhibited LR elongation. Could you give an explanation?

Response: In Arabidopsis, ethylene promotes LR formation and elongation. However, ACC had no significant effect on LR number but inhibited LR elongation in rice, indicating that different features are present in rice and Arabidopsis. To avoid misdirection, we deleted Line 175-179 in the revised manuscript.

Line 246-250: Again, it seems a bit contradictory… If ethylene mutants have a better penetration in compact soil, circumnutation should be important to it. If HK1 is a positive regulator of ethylene, how can this be useful for penetration in compact soils and root elongation?

Response: Ethylene promotes root circumnutation and inhibits root elongation in rice. Root circumnutation facilitates root growth past obstacles, whereas compacted soil inhibits root growth by restricting outward diffusion of ethylene through smaller air-filled soil pores. Thus, ethylene helps the roots to past obstacles but inhibits root elongation in harder soils, and precise manipulation of ethylene actions would be increase the performance of crops under specific agronomical conditions. We have explained this in the revised manuscript.

Reviewer 3 Report

Comments and Suggestions for Authors

Dear authors,
This review paper addresses the molecular mechanisms of ethylene in regulating root architecture and response to environmental signals. These mechanisms can contribute to genetic improvements or enhance adaptation to stressful environmental conditions.

The root system is essential in monocot crops such as rice, as the authors explain in the first part of the paper, how vital a good root system development is and the genes involved in this process.

The second part of the paper addresses the role of ethylene in root development, describing the signalling pathway and the interaction with Auxins in primary root formation. Additionally, the authors point out a need for more molecular evidence on crown-root and lateral-root development, especially their molecular mechanisms.

The third part of the paper addresses the regulation effects of ethylene in plants under abiotic stresses: salinity, drought, soil compaction, hypoxic conditions,  nutrients, and heavy metals. The authors explain how the ethylene signalling cascade interacts with other hormones, avoiding stress and improving root growth on genetically modified plants.

The paper's conclusions highlight ethylene's central role in root growth and downstream signal modulation by other hormones such as Auxines, ABA, JA, And GAs. Indeed, the role of ethylene in root development (especially primary root) has been extensively studied; however, as this paper explains, more studies are needed in lateral and crown root development. Breeding plants with a better root system is one way to improve crop yield, but a visualising system for phenotyping is needed. This subject could be the only thing this paper leaves out.

Finally, the number of references is more than adequate; Figure one is perfect, the table is self-explanatory, and Figure 2 is the only thing to change. I recommend publishing it as it is or with this minor correction.

Author Response

Many thanks for highlighting these strengths of our manuscript. We have modified Table 1 and Figure 2 as suggested by your and Reviewer 1.